

# Actors and networks in resource conflict resolution under climate change in rural Kenya

G. W. Ngaruiya[1] and J. Scheffran[2]

[1]     Department of Plant Sciences, Kenyatta University, Conservation Biology Section, Nairobi Kenya

[2]     Institute of Geography, University of Hamburg, Research Group "Climate Change and Security" Hamburg, Germany.

Correspondence to : G. W. Ngaruiya (ngaruiya.gracew@ku.ac.ke)

## Abstract

The change from consensual decision-making arrangements into centralised hierarchical chieftaincy schemes through colonization disrupted many rural conflict resolution mechanisms in Africa. In addition, climate change impacts on land use have introduced additional socio-ecological factors that complicate rural conflict dynamics. Despite the current urgent need for conflict-sensitive adaptation, resolution efficiency of these fused rural institutions has hardly been documented. In this context, we analyse the Loitoktok network for implemented resource conflict resolution structures and identify potential actors to guide conflict-sensitive adaptation. This is based on social network data and processes that are collected using the saturation sampling technique to analyse mechanisms of brokerage. We find that there are three different forms of fused conflict resolution arrangements that integrate traditional institutions and private investors in the community. To effectively implement conflict-sensitive adaptation, we recommend the extension officers, the council of elders, local chiefs and private investors as potential conduits of knowledge in rural areas. In conclusion, efficiency of these fused conflict resolution institutions is aided by the presence of holistic resource management policies and diversification in conflict resolution actors and networks.

## 1.  Introduction

Most African ethnic groups coexist peacefully with high degrees of mixing through inter-ethnic marriage, economic partnerships, and shared values that have been nurtured patiently over millennia(Aapengnuo, 2010). As a result, the management of



conflicts before colonization was guided by indigenous governance institutions that
established consensual decision-making arrangements at the grassroots (ECA, 2007).
This administrative role was later transferred to chieftaincies created by colonial
governments that sought to impose hierarchical rule on its subjects (Osaghae, 1989).
After independence, many African countries opted to maintain colonial administrative
structures and chieftaincy. To increase effectiveness in rural governance, local chiefs
were elevated to custodians of customary law and communal assets, with a
responsibility to dispense justice, resolve conflicts and enforce contracts (ECA, 2005).
This action created co-management regimes composed of diverse stakeholders,
representing divergent interests and interacting directly over a period of time to resolve
a specific conflict within their locality (Brunner et al., 2005).
However, the seemingly "stable" African conflict dialogue is currently becoming
complicated through additional socio-ecological factors from unpredictable climatic
conditions (Carius, 2009). There is a broad scientific debate whether and how climate
change may act as a 'threat-multiplier' and will increase resource conflicts in sub-Sahara
Africa(Lobell et al., 2008) among resource-dependent rural communities with low
adaptive capacity (AMCEN, 2011; Haldén, 2007; WRI et al., 2005). For clarity, a resource
conflict is defined in a wide sense as a situation whereby two or more parties
(individuals or groups) have or perceive to have, a) incompatible livelihood goals and
interests, or b) are in direct resource competition with each other and act upon these
differences (UNEP, 2009, 2011). Adaptation measures addressing impacts of climate
change on rural livelihoods have already been instituted globally to moderate potential
damages and/or exploit beneficial opportunities (IPCC, 2007).
But rigid demarcation into sectoral tasks of adaptation programmes can fall short
when it comes to conflict. Thus, a more systematic, integrated approach is necessary to
meaningfully incorporate existing conflict dynamics—as well as overarching socio-
political and economic conditions—into the design of adaptation measures. This creates
the need for conflict-sensitive adaptation strategies to enhance sustainable
development(Tänzler et al., 2013). Conflict-sensitivity refers to approaches and
measures that display cognisance of how: climate change can cause conflicts; climate
adaptation projects themselves can contribute to conflict and; adaptation measures
would operate in conflict zones(Yanda and Bronkhorst, 2011). Such knowledge allows
planners and decision-makers to address current vulnerabilities and development



priorities, while aiming to ensure long-term sustainability and peace through a basic
understanding of future projections(Yanda and Bronkhorst, 2011).
Consequently, this article seeks to address two knowledge gaps through this paper.
First, effectiveness of the fusion between indigenous mechanisms with conventional
and western conflict resolution approaches is still in question (ECA, 2007). Second, few
studies have documented actual rural structures and mechanisms used to resolve
resource conflicts in the sub-Saharan grassroots(Hyden et al., 2005). To this end, we
critically evaluate conflict resolution mechanisms of the water, agriculture and wildlife
sectors of Loitoktok district in Southern Kenya. We then use the brokerage concept
under social network analysis to identify central conflict resolution actors with the
potential to guide implementation of conflict-sensitive adaptation(Yanda and
Bronkhorst, 2011). We hypothesise that the presence of diverse stakeholders in the
conflict-resolution process contributes to high potential success in implementation of
conflict-sensitive adaptation in Loitoktok. Our discussion intends to further clarify local
conflict dynamics influencing adaptive capacity, social cohesion and rural development
in Kenya, as well as to contribute to the climate-security discourse in Africa.
The paper begins with a brief summary on the evolution of resource governance in
Africa. Then it elucidates capacity challenges of current rural resource governance in
dealing with potential climate-driven conflicts in sub-Saharan Africa and expounds on
the use of social network theory in diagnosing resource governance. Thirdly, a
description is given of the case study area of Loitoktok and the method used for
collecting and analysing social network data. The results and discussion are thereafter
presented based on identified conflict resolution mechanisms at the grassroots and
their potential in the climate adaptation discourse. A brief conclusion is given on key
highlights from the study.

## 92 2. Evolution of resource governance in Africa

Governance is defined as "the effective management of public affairs through the
generation of a regime (set of rules) accepted as legitimate, for the purpose of
promoting and enhancing societal values sought by individuals and groups"(Hyden et
al., 2005). It takes place through diverse institutions in a society, whereby, an institution
is likewise defined as an enduring collection of formal laws and informal rules, norms,



customs, codes of conduct, and organized practices that shape and govern human
interaction (IDRC, 2009).
African indigenous institutions of governance were altered radically with colonial
occupation that established a centralised governance system through the formalized
chieftaincy tactic that became the foundation of post-colonial governments of many
African countries (Cheka, 2008). After independence, the chieftaincy mandate was
further altered during fundamental restructuring of socioeconomic systems by African
political entities (ECA, 2007). Maintenance of the chieftaincy position was disputed by
some who were concerned with rapid growth and transformation of African economies.
For example, the late Tom Mboya quoted in (Osaghae, 1989) stated "Chieftaincy
impedes the pace of development as it reduces the relevance of the State in the area of
social services". Proponents of the chieftaincy stratagem highlighted differences
between the two systems that were clearly seen especially during conflict resolution, for
example the colonial (modern) legal system operates on the basis of an adversarial
approach while the traditional decision-making systems function on the basis of
consensual decision-making and reconciliation arrangements(ECA, 2007; IDRC, 2009;
IIDEA, 2011). Furthermore, since traditional institutions are indispensable for political
transformation in Africa, post-colonial governments opted to incorporate indigenous
knowledge into local administration regulations to increase positive perception of the
government by the masses(ECA, 2007).
Similarly, natural resources are embedded in a shared social space where complex
and unequal relations are established among a wide range of social actors, e.g. in the
case of the production of primary products, agro-export producers and farmers, ethnic
minorities, government agencies and others (Mwanika, 2010).However, the "one-size-
fits-all" governance approach introduced by colonialists gave  such poor outcomes that
led to the establishment of a rural participatory resource management approach to
promote community-based conservation, especially in developing countries (Berkes,
2004). The inclusion of indigenous  institutions and knowledge was important because
they guide how people negotiate access to resources and reduce (though not avoid
altogether) negative effects of conflict or drought (Eriksen and Lind, 2009). Apart from
indigenous institutions, many developing countries are currently implementing
poverty-reduction schemes that target the unemployed and marginalised groups. In
Kenya, the state has established among others the Revolving Fund for women and youth



community groups seeking to access business funds to improve their living standards (Ngaruiya and Scheffran, 2013).

Consequently, three main types of institutional governance systems are active in rural Africa.

a) *Traditional institutions* are defined as a power, permission or an institution emanating from indigenous authority that draws its legitimacy, whether wholly or partially, from tribal/ethnic/cultural values of a group of people that share them (Cheka, 2008). Such traditional institutions have either centralized or decentralized governance systems. Centralized systems had kings and monarchs such as the Abyssinia (Ethiopia), Buganda (Uganda) and Ashanti (Ghana) while decentralized systems comprise of  council of elders found among the Kikuyu and Maasai (Kenya), the gada (age-set) system of the Oromo in Ethiopia, or the Ibo village assembly in Nigeria (ECA, 2007).

b) *Formal institutions* are state-sponsored institutions that were mostly inherited from colonialism and constitute the written or codified rules such as the constitution, judiciary laws, organized markets, and property rights (IDRC, 2009; Mowo et al., 2013).

c) *Informal institutions* are the patterns of interdependence and actions among individuals who build themselves into different structural configurations to improve their living conditions or enhance resource exploitation. The actor linkages formed across the community vary by religion, ethnic identity, mode of production and are manifested as social networks (Prell et al., 2010).

When formal, informal and traditional institutions complement each other at different prefectures and different tiers, stakeholders are able to integrate diverse but relatable sources of knowledge to broaden resolution alternatives that might otherwise have been missed (Irwin et al., 2007). However, *institutional incoherence* is a major obstacle to effective governance. Incoherence occurs when governance institutions become incompatible to each other, with consequences such as hindrances in decision-making, wastage of financial resources or even deepened conflictual relationships at the grassroots (IDRC, 2009; Mowo et al., 2013). A practical incoherence example is seen in local adaptive capacity projects that are characterised by conflicting, overlapping mandates and dysfunctional arrangements in inter-agency integration as a result of weak coordination that subsequently gives poor outcomes (Madzwamuse, 2010). In



relation to this study, effective resource conflict governance calls for incorporation of
indigenous knowledge with a formal conflict resolution institution to create flexible
systems of resource management termed as "*adaptive co-management*" systems. These
systems become tailored to specific places and situations and are supported by and
work with various organizations at different levels (Folke et al., 2005). Furthermore,
effective adaptive co-management must involve multiple stakeholders to enhance
governance outcomes because local people know each other better, have more rapport
and sense of belonging that creates opportunities for cooperation and collective action,
for managing natural resources on a self-ruling and self-sufficient basis at the
grassroots (Mowo et al., 2013).
Against this background, issues of governance and institutional coherence turn out to
be relevant for our inquiry into the role of culture in social cohesion at the grassroots.

### 2.1. Capacity challenges in addressing climate-driven rural conflicts

Climate change has been described as a 'threat-multiplier' that may intensify existing
social, economic, political and environmental problems that communities are already
facing. Impacts of climate change are predicted to exacerbate grievances; overwhelm
coping capacities; and, in extreme times, spur forced or proactive migration(WBGU,
2008; Yanda and Bronkhorst, 2011).Climate change predictions for Africa suggest
increasing scarce water resources associated with declining and failing agricultural
yields in the Horn of Africa(Carius, 2009). Some studies predict a significant increase in
armed conflicts in sub-Saharan Africa by 2030 compared to the 1980 to 2000
period(Burke et al., 2009; Lobell et al., 2008)though others challenge this claim
(Buhaug, 2010; O'Loughlin et al., 2012). Various studies find mixed results on the
climate-conflict link in East Africa (Ide et al., 2014; O'Loughlin et al., 2012; Raleigh and
Kniveton, 2012; Schilling et al., 2012; Witsenburg and Adano, 2009).
Land is not just a material resource that people compete over, but it also forms the
basis of a particular way of life (farmer, pastoralist, fisher etc.); gives an ethnic identity;
and defines gender and age roles (Mwanika, 2010). Figure 1 illustrates possible paths to
conflicts induced by climate change in a typical rural village scenario in
Kenya(Ossenbrügge, 2009). These paths are termed as conflict constellations which are
divided into four, namely - water stress, food insecurity, drought as a natural disaster,
and migration issues (WBGU, 2008). Cumulative impacts from climate change on key



rural livelihood activities, such as agriculture and wildlife tourism, subsequently decrease (or cause failed) harvests and also increase farm raids by wildlife from neighbouring protected lands. Subsequent loss of income in rain-dependent communities lowers the spending power and increases local poverty levels. Affected households are thus left with land as their only asset which is viewed as an additional source of income, especially for rural households experiencing poor harvests and livestock productivity in Kenya (Ntiati, 2002). Subdivision of land disrupts the cultural norms and trusts of indigenous host communities through exposure to dissimilar immigrant norms and attitudes.

On the one hand, introduced norms could be beneficial like reduction of female genital mutilation. On the contrary, immigrants are perceived as "threats" who reduce power and influence of tribal chieftains, elites or local politicians. Such divisive thinking is grounded on the parochialism of communities in conceding the rights and interests of other communities (Western, 1994). On the extreme, if civic education is not foremost in the community then such a fragile "host vs. immigrant" situation creates fertile grounds for mobilizing citizens along ethnic or cultural lines by politicians vying for elective posts by promising "equal" resource allocation. Subsequently, people may retreat to their ethnic cocoons and agitate for social respite from the government. Such a "domino" effect clearly demonstrates the link between climate change impacts and resource conflicts whereby a decrease in ecosystem services production leads to increased rural poverty that gradually draws ever-deeper lines of division in social relations and triggers resource conflicts (WBGU, 2008). In the absence of conflict-sensitive adaptation programs, these resource conflicts become cyclic and reduce the ability of the community to adapt.

Despite that adaptation funding is already being made available and adaptation projects are under way in many rural communities (Yanda and Bronkhorst, 2011). Escalating cases of resource conflicts are projected to overwhelm rural conflict resolution mechanisms and reinforce the trend towards general instability and insecurity that already exists in many societies and regions (WBGU, 2008). We find that this prognosis is supported by three main capacity concerns drawn from the literature on climate change and resource conflicts in Africa:

a) (Handmer et al., 1999) posit that poorer regions and countries will have difficulty in adapting to climate change, since they lack comprehensive technical



and financial ability. In addition, African governments are faced with other major
developmental issues such as conflict, diseases and poverty that require direct
engagement by the state(AMCEN, 2011). Hence, at the moment climate change
adaptation policies seem unlikely to be successful or minimize inequity in Africa.
b) Adaptation is *not just* a technical process but also a political process since power
relations need to be adjusted for individuals and groups to achieve discrete
interests to maintain their own livelihoods (Eriksen and Lind, 2009). Poor
understanding of the African society structure and preference for "foreign" non-
governmental organisations (NGOs) with disparate interests in formulating the
African adaptation agenda has resulted in poor representation of the grassroots
level in the climate discourse, yet they are the most affected group (Hellmuth et
al., 2007; Madzwamuse, 2010).
c) Poor representation subsequently creates the third capacity challenge of
marginalisation of customary law in climate change policy-making at both
national and international levels, despite the high significance of indigenous
knowledge in the African society (AMCEN, 2011).Moreover, education systems
also neglect indigenous knowledge in school curricula concerning environmental
studies due to the negative undertone given to cultural practises by colonial
governments.
These three adaptive capacity issues infer that coherence between governance
institutions is critical in preventing competition over resources turning into a violent
conflict (Adano et al., 2012; Young, 2011). This is because effective adaptation can also
serve as a "threat minimiser" that brings together actors from security arrangements,
conflict resolution and asset management sectors to strengthen local adaptive capacity
while reducing predicted conflict cases (Donnelly-Roark et al., 2001). Furthermore,
opportunities for incorporating climate information into development activities in sub-
Saharan Africa are largely being missed at the moment (Hellmuth et al., 2007). This is
mainly because selecting representatives in resource governance institutions becomes a
complicated process since African rural communities are composed of diverse informal
interest groups that are formed as forums for exchanging knowledge, accessing
development funds and markets for their products (Ngaruiya and Scheffran, 2013).

**3. Use of social network theory in resource governance studies**



Incorporation of social network analysis into resource governance has rejuvenated
studies in natural resource management by introducing a quantitative approach to
political, economic or social processes in connection to structural and environmental
processes(Bodin and Prell, 2011). A social network is composed primarily of
interdependent actors together with the social relations (ties) linking these actors
together for transfer or flow of resources (Bodin and Prell, 2011). Social networks can
be viewed as a graph that consists of nodes (actors)joined by lines (relations) which
allows researchers to uncover patterns that might otherwise go undetected (Prell et al.,
2010). Network analysis fundamentally differs from standard social science research
because rather than focusing on attributes of autonomous individual units; it views
characteristics of the social units as arising out of structural or relational processes to
reveal theoretical motivations behind social relationships that shape environmental
outcomes(Wasserman and Faust, 1994).

Of interest to this study is how social network analysis facilitates

identification of stakeholder positions in a network and how these actors link various
parts of the network together (Bodin and Prell, 2011; Ngaruiya et al., 2015). Several
mathematical indices are used to quantitatively define this importance or prominence
of an individual actor within their social network. Equation (1) defines the *betweenness*
*centrality* index that counts the number of network pathways passing through an actor
and is used to measure how much potential control an actor has in disseminating
accurate and relevant information across the community network.
$$C_B(k) = \sum_{i \neq j \neq k} \frac{\partial_{ikj}}{\partial_{ij}} \qquad (1)$$
Where:

$C_B$(k) = betweenness centrality of actor k

$\partial_{ikj}$ = number of paths linking actors i and j that pass through actor k

$\partial_{ij}$ = number of paths linking actor i and j

This definition is based on the assumption that interactions between two
nonadjacent actors might depend on other actors, especially the actors who lie on the
path between the two (Wasserman and Faust, 1994). A practical implication of this
index is that if actors rest between many others, then they have the ability to "broker"
adaptation information to other actors and thereby influence the level of collective
knowledge in the community. If brokers are active within a community, they will not



only influence the quantity of knowledge but will also enhance the quality of knowledge circulating because they are able to connect diverse stakeholders to solve a common resource problem. For example, if a community has well-equipped brokers then the local ability to adapt to climate change increases the potential for peaceful conflict resolution and conflict transformation (Tänzler et al., 2013). On the other hand, unrestrained brokerage can create organisation chaos, manifest in errors such as resources allocated to conflicting goals and units in the same organisation competing against one another (Burt, 2011). A practical example of poor brokerage is how immense adaptation funding has caused a proliferation of actors offering diverse "expertise" in rural communities but with poor performance outcomes in many rural areas (Madzwamuse, 2010). Despite this flaw, brokerage is an interesting concept that is yet to be exhaustively applied in resource governance in Africa.

For that reason this paper uses social network analysis concepts to evaluate rural conflict resolution mechanisms, their structure and how central actors can be used to implement conflict-sensitive adaptation strategies at the grassroots.

## 4. Method

### 4.1. Area description

Our area of focus is Loitoktok district in Kajiado County, located at the southern tip of the former Rift Valley province in Kenya and covers c. 6,356.3 km². It is situated between longitudes 36º 5' and 37º5' East and between latitudes 1º0' and 3º0' South and borders the Republic of Tanzania to the West adjacent to Mt. Kilimanjaro (Government of Kenya, 2009). Ecologically, it is categorized among the arid and semi-arid districts in Kenya. The first census in 1962 showed a population of 24,027 persons while the current estimated population for 2012 is 171,520 persons. The district has an estimated annual population growth rate of 4.51% as per last census count (Government of Kenya, 2009).

Loitoktok was selected as representative of a typical Kenyan rural area because of a) its vibrant water, agriculture, and wildlife sectors, b) rapid land subdivision, c) introduction of diverse cultures by immigrants with different livelihood practises apart from pastoralism of the Maasai community. In addition, evidence of environmental impacts related to climate change have locally been documented through changes in precipitation (Thompson et al., 2009), temperature fluctuations (Altmann et al., 2002),



wildlife mortality (Wangai et al., 2013) and agricultural production (Ngaruiya 2014) in
Loitoktok.
In terms of governance institutions and stakeholder diversity, Loitoktok's rich
wildlife supports a strong tourism sector characterised by many hotels and lodges and
is rated as one of the key wildlife tourism areas in Kenya. Unmonitored land subdivision
and climate variability increased cases of wildlife poaching and human-wildlife conflicts
but these also created opportunities for establishment of several wildlife organizations
promoting conservation of local biodiversity. Additionally, due to the districts' remote
and semiarid location, several non-governmental organizations have been started to
boost the education, water and health sectors in collaboration with government
agencies.

**4.2. Data collection and analysis**
Field work was conducted in March-May and October-December 2012. Information
was sought on the resource conflict resolution process for water, wildlife and
agriculture sectors. A simple questionnaire collected relational (social network) data of
actor linkages using the saturation sampling technique within the Loitoktok community.
A respondent was asked to name five persons they share collaborations with during
conflict resolution and resource governance, whereby the named actors were located
(where possible) and asked to name their collaborators, which went on until no new
names were mentioned.
Thereafter, the social network data was converted into an actor matrix and analysed
for brokerage using the algorithm for betweenness centrality that finds the geodesics in
the network and then computes potential connections of every actor in the community.
The resultant data was then visualised as a sociograph using NetDraw™ that efficiently
illustrates the actual situation at the grassroots (Borgatti et al., 2002).

**5.  Results and discussion**
The respondents comprised of 152 persons drawn from four sectors (water,
agriculture, wildlife tourism and community) and also included expert interviews in
Nairobi, Kajiado and Loitoktok towns. The questionnaire also guided 6 group
discussions in Loitoktok All respondents agreed that inclusion of culture in the conflict
resolution process gave the community confidence in decisions agreed after





deliberations and that the main aim of a conflict resolution was to reduce tension or
violence by bringing the conflicting parties together. This coincides with principles of
natural resource management that emphasize the need for cooperation as a necessary
precondition for sustainable conflict resolution.

Table 1 illustrates practically how different resource conflicts were resolved between

November 2011 and November 2012 at Oloolopon Location in Loitoktok. It is evident
that resolving resource conflict is not the responsibility of a single person or institution,
but that minor conflicts were resolved by a small stakeholder meeting that was trusted
to recommend fair decisions for aggrieved parties, e.g. conflict over water at Impriron.
The most recommended discipline measure is compensation by the guilty actors to the
aggrieved party according to the level of destruction or damage. In extreme cases, when
the community felt aggrieved and the situation was thought to likely spread community
tension, the chief was obligated to call for joint meetings (*barazas)* for all relevant
stakeholders and the entire community.

This real-life reflection confirms that chiefs and other traditional authorities also

have the potential to mitigate ethnic conflicts by applying traditional conflict-resolution
mechanisms to narrow differences (ECA, 2007). The survey also confirms coherence
among the different institutions involved in resource conflict resolution.

### 5.1. Rural conflict resolution schemes

Three main conflict resolving systems borne from cooperative efforts were identified

in Loitoktok (Fig. 2). These are:-

a)  Policy-guided conflict resolution plan

Water scarcity was identified as a driver for resource conflict, especially during the dry
seasons in Loitoktok. As stipulated in the Water Act of 2002, the Ministry of Water
mandates its local government agency - Water Resources Management Authority
(WRMA) - to resolve local water conflicts together with the Water Resource Users
Association (WRUA). The local chief is an optional mediator in the presence of water
officials. If the conflict is not resolved through negotiation then it is either forwarded to
the courts for legal action against the offender or to the Water Appeals Board for further
arbitration. An interesting aspect is that WRMA also gives grants to approved WRUA's



projects that target enhanced water supply and quality. This clearly has encouraged the
community to participate in the prescribed regular training sessions for enhancing local
water governance.
Evidently, a well formulated resource policy is recognised as the first key step in
effectively resolving resource conflicts at the grassroots level. Thus the Water Act
clearly sets out the conflict resolution process and also empowers the resource users
with knowledge of their rights as resource users. For example, Loitoktok WRUA
members undertake citizen arrests of persons breaking water laws, especially upstream
farmers who over-extract water.

b)  Quasi-formal conflict resolution plan

This structural arrangement is predominantly used to solve two forms of conflict that
affect agricultural output. These are: i) Human-wildlife conflicts that occur when
wildlife invades farms for fodder or livestock (prey) and/or to access water sources. ii)
Farmer-pastoralist conflicts that occur when livestock destroy crops while trying to
access watering points since communal grazing areas have been lost following
subdivision of community group ranches. The agricultural conflict resolution
committee comprises of the formal council of elders (administrative type), the local
chief, agricultural extension officers and police. This arrangement is termed as quasi-
formal because the elders and chief are nominated from the community by the
government, unlike in the water sector that only works with civil servants in conflict
resolution. The committee uses a crop damage or livestock death report prepared by
the extension officer to guide negotiations after which the aggrieved party is
compensated either in kind (livestock) or in cash form. Police is involved to ensure that
the conflict resolution process can be transferred to court if the offender fails to fulfil
the stipulated compensation. Though the council of elders is part of the community
sometimes the community perceives their unfavourable rulings with suspicion as if they
represent the government.

c)  Hybrid site-specific conflict resolution plan

The wildlife sector exhibits a unique conflict resolution strategy as a result of
inadequate government policies. This strategy comprises of the traditional council of
elders, formal government agencies, private investors and researchers who come



together tocover shortcomings of the wildlife conflict management strategy. For
example, previous absence of compensation for livestock deaths and crop destruction
by wildlife led to wanton slaughter of lions, elephants, or zebras. Now, modest payments
to aggrieved families by private investors such as Mr Luke of Olkeri Sanctuary for losses
incurred by predators or elephants have reduced cases of revenge wildlife killings.
Another example was seen at the Mbirikani group ranch whereby game scouts
(members of the community) conduct regular patrols. Respondents stated that since the
*community wildlife policing* project has begun, the poaching levels have been reduced.
This site-specific measure infers that community members are prone to cohesively use
their own knowledge if they are assisted in developing an efficient way of collaboration
to enhance their livelihoods.
Though this mechanism seems to be effective, it does not comprehensively deal with the
fundamental cause of human-wildlife conflicts. This is because no policy exists on how
to ensure survival of wildlife during drought episodes to prevent the recurrent human-
wildlife conflicts. In addition, poor understanding of the wildlife management policy has
also exposed the community to manipulation by politicians seeking voter mileage at the
expense of the human-wildlife incidents.
From this study, it is evident that the resource conflict resolution process involves
interaction among diverse actors which in turn increases local civic knowledge,
community participation and shows respect to cultural practices that together
strengthen rural community networks. This also signals effectiveness of the fusion
between indigenous and conventional conflict resolution mechanisms. Furthermore,
integration of diverse stakeholders provides a basis to broaden institutional networks
and partnerships through alternative livelihood activities that may boost the local
economy. However the need to overhaul the land policy in Kenya cannot be overlooked.
Respondents stated that a comprehensively developed land policy will establish zones
for different development purposes and allocate buffer zones to reduce incidences of
encroachment and human-wildlife conflict in protected areas. This action will ensure
that future urban expansion will not lead to resource competition or unequal
distribution in rural areas of Kenya.

**5.2. Loitoktok social governance structure**



Scrutiny of the resource governance and conflict resolution structures reveal 86 actors in 23 formal institutions (government agencies), 16 informal institutions (community groups), 46 private organizations and 1 traditional institution. These institutions belong to four main sectors namely, agriculture, wildlife & forestry, water resources and community management that implement resource governance through collaborative actions from 30, 31, 11 and 14 actors from the respective sectors.

Figure 3 gives an illustration of how actors are connected and also identifies actors who occupy the central position in Loitoktok. Full names of actors are contained in the supplement. These actors are more visible, have the highest degree of ties and are involved centrally in resource conflict resolution in the network. They include: District Agricultural Officer (DAO), District Kenya Wildlife Service (DKWS), District Livestock Officer (DLO), District Local Government (DLG), Social Development Officer (DSDO), District Water Officer (DWO), District Kenya Forest Service (DKFS) and game scouts. The calculated betweenness scores that indicate the network influence of the identified central actors are 718.5, 670.5, 179.5, 165, 151, 80, 78 and 78, in the same order respectively. These values represent currently missing links to neighbouring actors that can be potential links available for each actor to use in increasing the number of connection in the network. The eight actors have the highest ability to build resource knowledge and ecosystem dynamics so that the community can collectively respond to environmental feedback in a fashion that contributes to resilience. The rest of the actors have betweenness scores of less than 10 and thus have a small effect on information dissemination and control within the larger community.

By empowering the central actors to actively create connections that span across different resource sectors then the community can strengthen the local governance strategy for effective problem-focused community resource management. This is discussed below.

### 5.3. Building conflict-sensitive adaptation

Conflict resolution is critical to adaptation as conflict restricts many drought adjustments involving peaceful interaction between many diverse stakeholders. Conflict-sensitive adaptation becomes therefore a holistic, multi-scaled and multi-sectored approach that taps into the wealth of traditional knowledge regarding the management of resources and conflicts at a community level(Yanda and Bronkhorst,



2011).Since conflict-sensitive adaptation processes must be approached using a multi-
dimensional system that incorporates different levels, both administrative and societal
(Tänzler et al., 2013). Then, this study postulates that central actors, who hold the
network together in times of distress, also have potential to influence adaptation
information quality and flow in the network.

Loitoktok actors who should be equipped with adaptation knowledge to "broker" to

the community are:

a)    Extension officers

These are the District Agricultural Officer (DAO), District Livestock Officer (DLO),
District Kenya Forestry Service (DKFS) and District Kenya Wildlife Service (DKWS). The
extension officers are well connected to their respective community interest groups
(informal institutions) and thus can be effective in transfer of adaption knowledge. The
community indicated that water and wildlife sectors recorded the highest number of
conflicts and subsequent studies have confirmed low adaptation measures in these two
sectors. Conversely, crop and livestock sectors have the most diverse adaptation
measures due to a close public-private actor partnership (Ngaruiya, 2014). Therefore,
specialised training of extension officers in adaptation technology and water harvesting
for subsequent transfer to the community will not only buffer food security (crop and
livestock products) but will also strengthen the local economy through creation of
additional livelihood opportunities in a climate change context.

b)    Council of elders

In Loitoktok, the outstanding traditional institution is the Council of Elders that is

made up of persons of integrity and objectivity who have distinguished themselves in
one way or another and have been recognized as such by the community (Cheka, 2008).
There are two types of Council of Elders. First, the Council of Elders that is appointed by
the State and is made up of men from the three major tribes in the district to help in
administration issues such as immigration and conflict resolution in the agriculture
sector (quasi-formal). Secondly, the dominant host Maasai community exclusively
selects its indigenous Maasai Council of Elders (traditional institution) according to its
culture which is also respected by other communities in Loitoktok. This council is highly
regarded in the wildlife sector where it plays a key role in either agitating for action by
the government and investors or calming the Maasai community after a serious human-



wildlife incident. Interestingly from the social network analysis, the council of elders is
not among the top central actors because of the administrative dichotomy in the district.
But the fact still remains that they are well connected to each resource sector, thereby
giving them a stronger knowledge dissemination power in the community.

In terms of judgements and costs, indigenous conflict resolution mechanisms have
been found to be effective for both lesser criminal cases such as stock theft, land
disputes and serious crimes such as genocide as seen in Rwanda (ECA, 2007). Hence
incorporating such respected institutions originating from customary law and
indigenous knowledge into climate change policies is likely to result in formulation of
effective adaptation strategies that will be participatory and highly acceptable by the
rest of the community.

c)    Local chief

Loitoktok has 16 locations each governed by a chief and 31 sub-chiefs who are in-
charge of sub-locations. These chieftaincy positions are not elective but the person is
nominated by the government to participate in decision-making at the grassroots. The
chiefs work under the District Local Government (DLG) office and are called upon by the
government depending on the conflict situation in the community. The administrative
council of elders also falls under the DLG office as a physical representation of the
government in the community. These quasi-formal arrangements are alternative
institutions that are peripherally involved in resource governance but can also improve
the climate change discourse in Africa. The chiefs and council of elders can identify
isolated rural community interest groups for training in resource governance including
conflict resolution since unmanaged informal groups form many small and dense
clusters with little or no diversity and little adaptation knowledge that become resistant
to change. An example is pastoralists who view livestock as a form of wealth and calls by
extension officers  to dispose of healthy animals before onset of drought is viewed with
suspicion. Furthermore, chiefs can conduct civic lessons among their constituents as a
means of promoting integration and coexistence and dispelling false information to
foster the concept of "a common people with a common destiny" (Aapengnuo, 2010).

557        d)    Private investors and researchers



Loitoktok network has many private organisations such as hotel owners, seed
companies' researchers, humanitarian workers etc. in all the resource sectors. Most
private actors are seen to be more effective in resolving conflicts in the wildlife sector as
a way of preserving the wildlife resource that attracts tourists to the area. Societal
decision-making is nested in a wider set of societal changes, such as institutional
changes and altered relations between public and private actors. Thus, for a community
to increase its adaptive capacity then it should incorporate all stakeholders in
developing land and resource management designs to make them more effective and
relevant to investors. Apart from formal institutions and the non-governmental
organisations, communities should incorporate local investors who have financial and
technical ability to support the community in sustainable use of biodiversity and
practical knowledge to maintain ecosystems in good condition to avoid conflicts over
scarce resources especially during drought.

## 6. Conclusions

A number of studies have used economic, political and ecological aspects to expound
resource conflicts in several African countries. However, few studies have documented
the social structures that resolve conflicts at the grassroots. This study confirms that in
post-colonial Kenya, resource governance still contains vestiges of traditional
institutions, especially in collective discussion of grievance towards effective conflict
resolution. The innovative arrangements make use of indigenous knowledge to calm the
aggrieved and agitate for compensation by the government. As a result, this integration
binds the society together by its inherent customs based on brotherhood notions for
enhanced resource utilisation and livelihoods, regardless of climatic conditions.
Secondly, climate change threatens to disrupt conflict resolution mechanisms that are
operational in rural centres because of capacity challenges associated with Africa's low
technical ability to manage climate governance, poor integration of diverse opinions
and marginalisation of indigenous knowledge into adaptation and mitigation agendas.
We based the field study on the Loitoktok district that is expanding in terms of its
cultural diversity, economic sectors and profile of resource conflict which is
representative of many rural areas in Kenya as well as sub-Saharan countries. Results
indicate that conflict resolution was achieved through three forms of institutions, each
unique to its natural resource. The water sector relied upon its comprehensive policy;



agriculture used a quasi-formal arrangement while the wildlife sector formulated its
own hybrid arrangement that involved private investors and the traditional council of
Maasai elders. In extreme cases, the community came together in *barazas* to air their
concerns and agree on a collective decision acceptable to all relevant stakeholders.
Implementation of conflict-sensitive adaptation requires a deep understanding of the
context in which climate-driven resource-conflicts are resolved in a community and
clearly delineated actor interactions between local resource-related activities.
Therefore, we used the betweenness centrality index drawn from the flourishing field of
social network theory to evaluate the central actors with potential to broker adaptation
knowledge across the Loitoktok network. Results indicate that extension officers,
council of elders, local chief and private investors are the suitable central actors who
should be financially and technologically equipped for building conflict-sensitive
adaptive capacity in the community. Thus government and non-government
stakeholders must work together to identify risks and formulate strategies and
programmes that can help raise awareness among civil society of the impact of climate
change.
As a contribution to the climate and security discourse, this study advocates for two
adaptive co-management measures to help overcome climate change-related capacity
challenges at the grassroots in Africa. First, clear conflict resolution policy in natural
resource governance as seen in the water sector will help solve local conflicts and also
enable stakeholders to understand local conflict genesis and effectively prepare for
unpredictable climatic conditions. Secondly, involving diverse actors from the
community in resolving conflict as seen in the wildlife sector, also has potential in
serving as a conduit of the adaptation knowledge sector that empower the community
despite policy inadequacies. Moreover, traditional institutions like the council of elders
have been seen as a source of civic knowledge, and encourage respect of local values
and customs that contribute to community self-reliance and empowerment in the
community.
To conclude, natural resource regulations and governance arrangements play
important roles in handling potential conflicts over scarce natural resources,
particularly water in arid and semi-arid lands (ASAL). Thus resource conflict resolution
and positive culture transmission should be part of an effective conflict-sensitive
adaptation strategy. These two aspects encourage growth of cohesive social capital that



in turn enhances economic development at the grassroots and effective governance of
the commons.



**Acknowledgments**
Special thanks go to Mr. Gadayo (Wiper), Prof. John W. Kiringe and Mr. Paul Ngugi
(Treasury) for assistance in Kenya. This work is supported by
DeutscherAkademischerAustauschdienst (DAAD), National Council for Science and
Technology (NCST)-Kenya, German Science Foundation (DFG) through Cluster of
Excellence "Integrated Climate System Analysis and Prediction", and School of
Integrated Climate System Sciences (SICSS) as a PhD project at the University of
Hamburg, Germany.

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



Table 1. The annual resource conflict report of Chief Leonard Kasine in-charge of
Oloolopon Location in Loitoktok district. (WRMA: Water Resources Management
Authority, KWS: Kenya Wildlife Service).

| Resource | No. of conflicts | Conflict site | Resolution | Stakeholders involved |
|---|---|---|---|---|
| Water | 3 | Shurie | Compensation | Council of elders, Chief and residents |
| | 7 | Impiron | Community discussion | WRMA and Chief |
| | 1 | Airstrip | Community discussion | Nolturesh Water Board and Chief |
| Livestock | 16 | Korinko village | Fine after agricultural assessment | Agricultural extension officers, police, Chief |
| | 26 | Inkariak-Rongena | 4 fined by court 22 fined after agricultural assessment | Agricultural extension officers, police, Chief |
| | 11 | Kamukunji | Compensation to farmer | Agricultural extension officers, Chief |
| Wildlife | 30 | Sompet | Compensation | KWS, Private investor – Elephant Research Org. |
| | 6 | Ilmisigiyio | Compensation | KWS, African Wildlife Foundation |






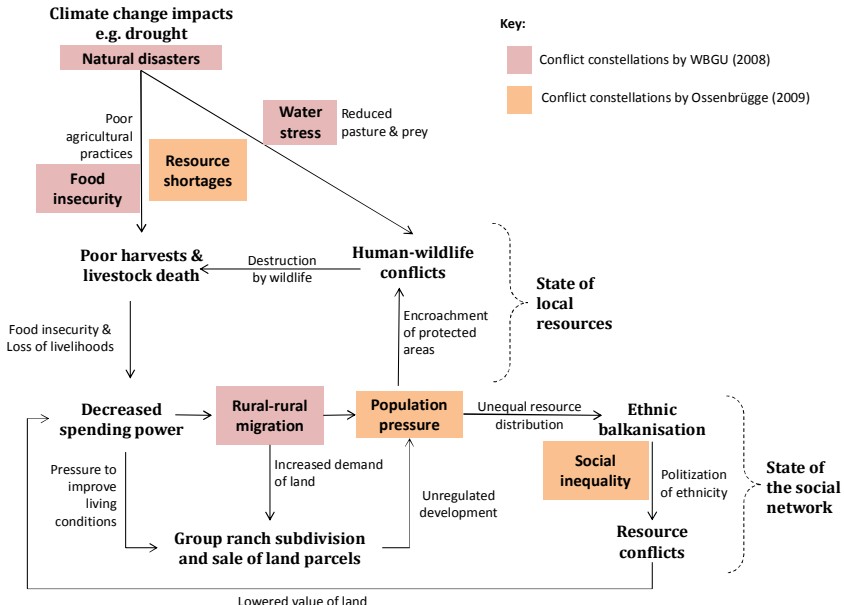

Figure 1. Conflict constellations in relation to climate change and rural land tenure.
Source: The authors.



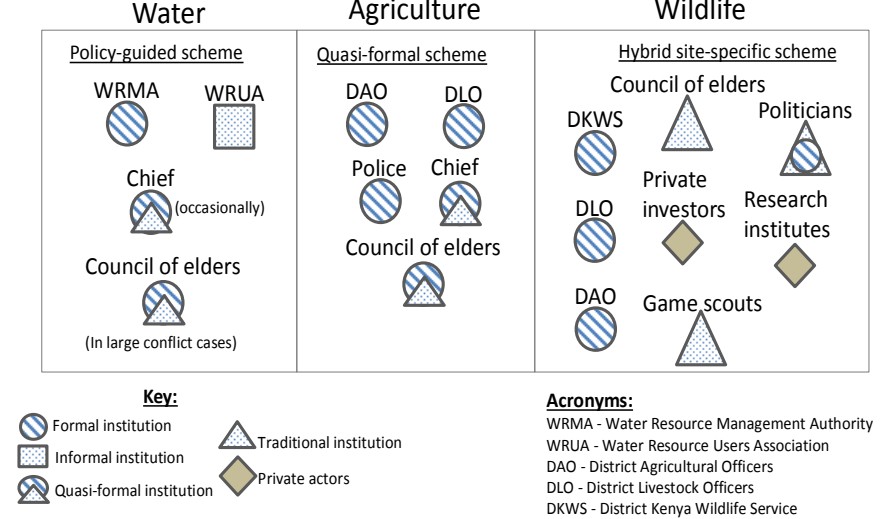


Figure 2. The diverse resource conflict resolution schemes in Loitoktok district.





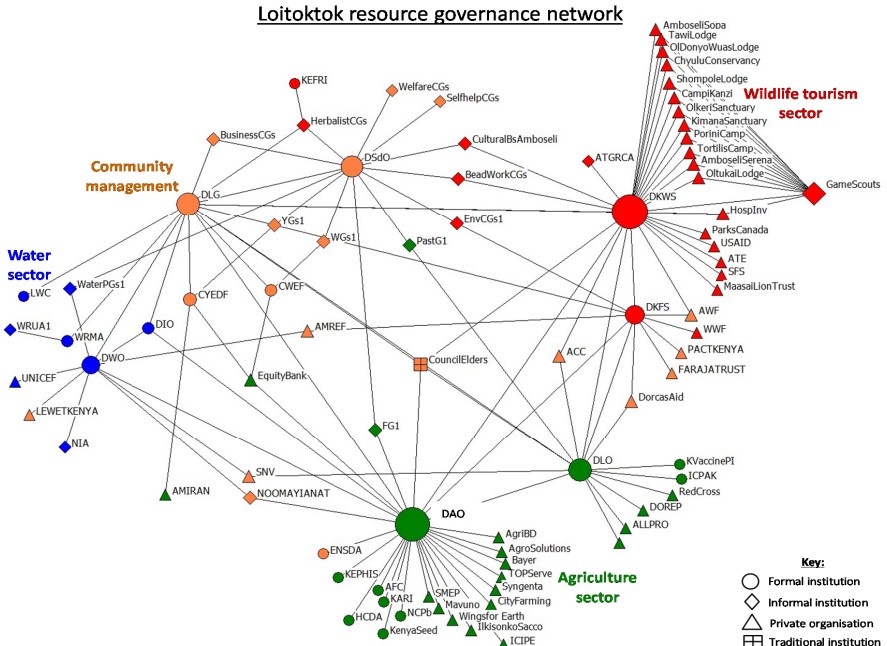


Figure 3. Social network illustrating actor linkages in resource governance at Loitoktok
community.