# Peer review of "Actors and networks in resource conflict resolution under climate change in rural Kenya"

_Earth System Dynamics, 2015_

## Referee Comment (RC1) · J. Njoroge (Referee) · 14 Feb 2016

The manuscript is relevant to the scope of the journal and i find it interesting. The manuscript articulates important issues on conflict resolution on matters subject to climate change. Climate change related conflicts ought to continue among vulnerable societies and understanding how different societies are able to handle their conflict resolution through either traditional or modern approaches through case studies such like this will further inform future policies. It also has a good background on traditional conflict resolution in Africa. In order to enhance this manuscript i have a number of suggestions: 1. Line 12 change the word 'into' to 'to' 2. Statement 56-55 'Adaptation measures...' is misplaced in the paragraph or rather lacks connection to the ideas ar-

ticulated in the paragraph. It fits better as an opening statement for the next paragraph 3. Statement on line 221 is incomplete. 4. It will be of significance to cite few studies that have applied your methodology and or that have focused on your theme. 5. Finally in your conclusion line 573 you have mentioned and affirmed about existence of few studies. Are you able to cite some of them in order to guide further reading? 6. Further more you need to guide your discussion in relation to other studies in the region or focusing on the theme. 7. Also line 574-575 statement also needs affirmation by providing examples of those studies. 8. It would also be of interest to have your opinion on how formal judicial mediation and traditional approaches may be strengthened through collaborative efforts. Just a thought

————————————————

---

## Referee Comment (RC2) · Anonymous Referee #2 · 20 Feb 2016

General Comments Climate change impacts on many Kenya rural communities have had severe consequences including intensification of inter and intra community conflicts. Although there has been a lot of discussion on adaptation tactics particularly Community based Adaptation (CBA) and Ecosystem based Adaptation (EbA), little attention has been paid to the role of community networks in helping rural communities adapt to climate change impacts. I found this paper very interesting and informative and I believe it will trigger more discussions in the field of adaptation to climate change. The paper is also very well written in fluent language and also structured appropriately. However, I have a few suggestions which if implements can improve the paper's quality.

Specific comments 1. Line 100 on page 4. I suggest replacing "radically with colonial"

with "radically by colonial" 2. Page 4 lines 120 – 125.It will be important to include references/examples of site 3. s where poor outcomes have been recorded. 4. Pages 7, lines 201 to 205. It is implied that once land becomes the only economic asset remaining, the land owner sub-divides for sale to "outsiders". This needs to be clarified. 5. The fact that sale of land to "outsiders" leads to reduced land holdings, reduced grazing area, increased incidences of overgrazing, fencing – all these lead to reduced adaptive capacity. This needs to be highlighted. 6. Page 11. Lines 332. Sentence starting on line 332 needs to be broken. I suggest "Unmonitored land subdivision and climate variability, increased cases of wildlife poaching and human-wildlife conflicts......" 7. Page 15: lines 469 to 472. Replace "District Agricultural Officer (DAO), District Kenya Wildlife Service (DKWS), District Livestock Officer (DLO), District Local Government (DLG), Social Development Officer (DSDO), District Water Officer (DWO), District Kenya Forest Service (DKFS) and game scouts" with "District Agricultural Officer (DAO), District Kenya Wildlife Service (DKWS) officer, District Livestock Officer (DLO) Officer, District Local Government (DLG) officer Social Development Officer (DSDO), District Water Officer (DWO), District Kenya Forest Service (DKFS)officer and game scouts. 8. You also need to confirm the titles for the officers since the title District is no longer applicable – we have counties and sub-counties. There is need to confirm this and change throughout the paper. 9. Page 16 lines 493 – 497 "Since conflict-sensitive adaptation processes must be approached using a multi dimensional system that incorporates different levels, both administrative and societal (Tänzler et al., 2013). Then, this study postulates that central actors, who hold the network together in times of distress, also have potential to influence adaptation information quality and flow in the network." Needs rephrasing. I suggest "Conflict-sensitive adaptation processes must be approached using a multi dimensional system that incorporates different levels, both administrative and societal (Tänzler et al., 2013). This study postulates that central actors, who hold the network together in times of distress, also have potential to influence adaptation information quality and flow in the network." 10. Page 16 – lines 501 – 502 and also ather oparts of the paper – We no longer have districts in

Kenya but Counties and Sub-counties. The authors needs to find out Loitoktok is now a sub-county.

---

## Author Comment (AC2) · 29 Mar 2016

We would like to appreciate comments from the anonymous referee who highlighted the need to change the administrative titles of the actors to reflect the current Kenyan Constitution that created Counties and Sub-Counties instead of Districts.

In addition, the other suggested specific comments guided our reply and changes to the manuscript as follows;

1. Line 100 on page 4. I suggest replacing "radically with colonial" with "radically by colonial" – accepted change.

2. Page 4 lines 120 – 125.It will be important to include references/examples of sites

where poor outcomes have been recorded – Accepted review. For clarity, we have edited the sentence as "However, the "one-size-fits-all" governance approach introduced by colonialists gave poor outcomes especially in the water, wildlife and forest sectors, thereby necessitating establishment of rural community-based conservation and participatory resource management approaches in developing countries (Berkes, 2004)."

3. Pages 7, lines 201 to 205. It is implied that once land becomes the only economic asset remaining, the land owner sub-divides for sale to "outsiders". This needs to be clarified - The statement is given in relation to figure 1 which assumes that majority of the community becomes impoverished due to local drought impacts and the "outsiders" have higher capability of enhancing the "locals" economic status. Consequently, the sentence has been changed to clarify the circumstance under which land is sold to outsiders.

4. The fact that sale of land to "outsiders" leads to reduced land holdings, reduced grazing area, increased incidences of overgrazing, fencing – all these lead to reduced adaptive capacity. This needs to be highlighted. Accepted idea as it emphasizes the need for effective holistic adaptation practices in a community. This suggested sentence has been included in the paragraph.

5. Page 11. Lines 332. Sentence starting on line 332 needs to be broken. I suggest "Unmonitored land subdivision and climate variability, increased cases of wildlife poaching and human-wildlife conflicts. . ...." Accepted change, the sentence now reads "Interestingly, unmonitored land subdivision and climate variability, increased cases of wildlife poaching and human-wildlife conflicts but have also created opportunities for establishment of several wildlife organizations that promote conservation of local biodiversity"

6. Page 15: lines 469 to 472. Replace "District Agricultural Officer (DAO), District Kenya Wildlife Service (DKWS), District Livestock Officer (DLO), District Local Government (DLG), Social Development Officer (DSDO), District Water Officer (DWO), District Kenya Forest Service (DKFS) and game scouts" with "District Agricultural Officer (DAO), District Kenya Wildlife Service (DKWS) officer, District Livestock Officer (DLO) Officer, District Local Government (DLG) officer Social Development Officer (DSDO), District Water Officer (DWO), District Kenya Forest Service (DKFS)officer and game scouts. Accepted, I have included the term officer in the actors.

7. You also need to confirm the titles for the officers since the title District is no longer applicable – we have counties and sub-counties. There is need to confirm this and change throughout the paper. Accepted change from District to Sub-County officers (SCO) to concur with current government administrative structures.

8. Page 16 lines 493 – 497 "Since conflict sensitive adaptation processes must be approached using a multi-dimensional system that incorporates different levels, both administrative and societal (Tänzler et al., 2013). Then, this study postulates that central actors, who hold the network together in times of distress, also have potential to influence adaptation information quality and flow in the network." Needs rephrasing. I suggest "Conflict-sensitive adaptation processes must be approached using a multi-dimensional system that incorporates different levels, both administrative and societal (Tänzler et al., 2013). This study postulates that central actors, who hold the network together in times of distress, also have potential to influence adaptation information quality and flow in the network." Accepted sentence flow change.

9. Page 16 – lines 501 – 502 and also other parts of the paper – We no longer have districts in Kenya but Counties and Sub-counties. The authors' needs to find out Loitoktok is now a sub-county. Accepted change from Loitoktok District to Sub-County to correspond with the current government administrative unit.

---

## Author Comment (AC3) · 29 Mar 2016

We have changed the names of actors from District to Sub-County to correspond to current administrative units in Kenya. These changes are both in the manuscript, figures and the supplement as noted below.

Please also note the supplement to this comment:
http://www.earth-syst-dynam-discuss.net/esd-2015-89/esd-2015-89-AC3-supplement.pdf
* * *
[Figure]

**Fig. 1.** The diverse resource conflict resolution schemes in Loitoktok Sub-County.

**Loitoktok resource governance network**

**COMMUNITY MANAGEMENT**

BusinessCGs
WelfareCGs
SelfhelpCGs
SCG
WGs1
SDO
CulturalBsAmboseli
BeadWorkCGs
YGs1
EnvCGs1
CYEDF
HerbalistCGs
CWEF
AMREF
FG1
CouncilElders
SNV
HospInv
WRMA
EquityBank
SCAO
PastG1
AWF
LWC
WaterPGs1
SCIO
NOOMAYIANAT
SMEP
SCLO
ACC
WRUA1
AFC
KARI
Syngenta
DorcasAid
SCWO
ICIPE
AgroSolutions
DOREP
NIA
HCDA
Bayer
CityFarming
RedCross
ENSDA
KEPHIS
AgriBD
Mavuno
ALLPRO
UNICEF
NCPb
Wingsfor Earth
SACDEP
**WATER SECTOR**
LEWETKENYA
AMIRAN
IlkisonkoSacco
**AGRICULTURE SECTOR**
ICPAK
KenyaSeed
TOPServe
KVaccinePI

OlDonvoWuasLodge
KimanaSanctuary
CampiKanzi
OlkeriSanctuary
AmboseliSopa
ChyuluConservancy
OltukaiLodge
**WILDLIFE TOURISM SECTOR**
PoriniCamp
ShompoleLodge
TortilisCamp
TawiLodge
AmboseliSerena
Tourists
SCKWS
GameScouts
ATE
SFS
ParksCanada
MaasaiLionTrust
USAID
ATGRCA
SCKFS
WWF
PACTKENYA
FARAJATRUST
KEFRI

Key:
○ Formal institution
◇ Informal institution
△ Private organisation
⊞ Traditional institution

[revised manuscript text omitted]

---

## Author Response (AR1)

* * *
**Referee # 1 J. Njoroge** joseph.muiruri@hotmail.com

General Comments
The manuscript is relevant to the scope of the journal and I find it interesting. The manuscript articulates important issues on conflict resolution on matters subject to climate change. Climate change related conflicts ought to continue among vulnerable societies and understanding how different societies are able to handle their conflict resolution through either traditional or modern approaches through case studies such like this will further inform future policies. It also has a good background on traditional conflict resolution in Africa.

Specific comments
In order to enhance this manuscript I have a number of suggestions:

1. Line 12 change the word 'into' to 'to' – Accepted change

2. Statement 56-55 'Adaptation measures...' is misplaced in the paragraph or rather lacks connection to the ideas articulated in the paragraph. It fits better as an opening statement for the next paragraph – Accepted change as a new sentence in the next paragraph

3. Statement on line 221 is incomplete – correction on the sentence to read "Adaptation funding is already being made available and adaptation projects are under way in many rural communities (Yanda and Bronkhorst, 2011). However, escalating cases of resource conflicts are projected to overwhelm rural conflict resolution mechanisms and reinforce the trend towards general instability and insecurity that already exists in many societies and regions (WBGU, 2008)."

4. It will be of significance to cite few studies that have applied your methodology and or that have focused on your theme.  Several literatures are quoted in the manuscript that reflects:
   I. *The social network analysis methodology e.g.*
      a. Bodin, Ö. and Prell, C.: Social Networks and Natural Resource Management: Uncovering the Social Fabric of Environmental Governance, Cambridge University Press., 2011.

b. Burt, R.: Brokerage and closure: an introduction to social capital, Oxford University Press, Oxford; New York., 2011

c. Wasserman, S. and Faust, K.: Social network analysis methods and applications, Cambridge University Press, Cambridge; New York., 1994.

II. *The theme of climate change and resource conflicts in sub-Sahara e.g.*

a) Buhaug, H.: Climate not to blame for African civil wars, Proc. Natl. Acad. Sci., 107(38), 16477–16482, 2010.

b) Burke, M. B., Miguel, E., Satyanath, S., Dykema, J. A. and Lobell, D. B.: Warming increases the risk of civil war in Africa, Proc. Natl. Acad. Sci., 106(49), 20670–20674, 2009.

c) Carius, A.: Climate Change and Security in Africa Challenges and international policy context, United Nations, Berlin, Germany. 2009

5. Finally in your conclusion line 573 you have mentioned and affirmed about existence of few studies. Are you able to cite some of them in order to guide further reading? I have included a short reference in the paragraph indicating where the referred literature can be found i.e. the literature cited in section 2.1. in lines 178 – 189 that discuss capacity challenges in climate driven resource conflicts.

6. Furthermore you need to guide your discussion in relation to other studies in the region or focusing on the theme. The issue of resource conflicts in relation to climate change is addressed in section 2.1., especially in lines 249 – 260 where we highlight the knowledge gap that we seek to address.

7. Also line 574-575 statement also needs affirmation by providing examples of those studies. This sentence in the conclusion section reflects the studies cited within the text namely: Hyden et al., 2005; Eriksen and Lind, 2009 and, Ngaruiya and Scheffran, 2013 in lines 72, 236 and 270 respectively.

8. It would also be of interest to have your opinion on how formal judicial mediation and traditional approaches may be strengthened through collaborative efforts. Just a thought! Reflecting on the study, we propose that the government formulates holistic resource management policies that institutionalize hybrid governance structures down to the local level. This action will provide a forum for; 1) The community to incorporate indigenous knowledge into management of their natural resources, promote cultural knowledge transmission and resolve resource conflicts using traditional structures that are significant to the community. 2) The forum will also help growth of social capital which will aid increase in adaptive capacity and enhance local economic development.

**Anonymous Referee #2**

General Comments

Climate change impacts on many Kenya rural communities have had severe consequences including intensification of inter and intra community conflicts. Although there has been a lot of discussion on adaptation tactics particularly Community based Adaptation (CBA) and Ecosystem based Adaptation (EbA), little attention has been paid to the role of community networks in helping rural communities adapt to climate change impacts. I found this paper very interesting and informative and I believe it will trigger more discussions in the field of adaptation to climate change. The paper is also very well written in fluent language and also structured appropriately.

However, I have a few suggestions which if implements can improve the paper's quality.

Specific comments

1. Line 100 on page 4. I suggest replacing "radically with colonial" with "radically by colonial" – accepted change.

2. Page 4 lines 120 – 125.It will be important to include references/examples of sites where poor outcomes have been recorded – Accepted.  For clarity, we have edited the sentence as "However, the "one-size-fits-all" governance approach introduced by colonialists gave poor outcomes especially in the water, wildlife and forest sectors, thereby necessitating establishment of rural community-based conservation and participatory resource management approaches in developing countries (Berkes, 2004)."

3. Pages 7, lines 201 to 205. It is implied that once land becomes the only economic asset remaining, the land owner sub-divides for sale to "outsiders". This needs to be clarified. The statement is given in relation to figure 1 which assumes that majority of the community becomes impoverished due to local drought impacts and the "outsiders" have higher capability of enhancing the "locals" economic status. The wording has been changed to clarify.

4. The fact that sale of land to "outsiders" leads to reduced land holdings, reduced grazing area, increased incidences of overgrazing, fencing – all these lead to reduced adaptive capacity. This needs to be highlighted. Accepted idea as it emphasises the need for effective holistic adaptation practices in a community.

5. Page 11. Lines 332. Sentence starting on line 332 needs to be broken. I suggest "Unmonitored land subdivision and climate variability, increased cases of wildlife poaching and human-wildlife conflicts. . ...." Accepted change, the sentence now reads "Interestingly, unmonitored land subdivision and climate variability, increased cases of wildlife poaching and human-wildlife conflicts but have also created opportunities for establishment of several wildlife organizations that promote conservation of local biodiversity"

6. Page 15: lines 469 to 472. Replace "District Agricultural Officer (DAO), District Kenya Wildlife Service (DKWS), District Livestock Officer (DLO), District Local Government (DLG), Social Development Officer (DSDO), District Water Officer (DWO), District Kenya Forest Service (DKFS) and game scouts" with "District Agricultural Officer (DAO), District Kenya Wildlife Service (DKWS) officer, District Livestock Officer (DLO) Officer, District Local Government (DLG) officer Social Development Officer (DSDO), District Water Officer (DWO), District Kenya Forest Service (DKFS)officer and game scouts. Accepted, I have included the term officer in the actors.

7. You also need to confirm the titles for the officers since the title District is no longer applicable – we have counties and sub-counties. There is need to confirm this and change throughout the paper. Accepted change from District to Sub-County officers (SCO) to concur with current government administrative structures.

8. Page 16 lines 493 – 497 "Since conflict sensitive adaptation processes must be approached using a multi-dimensional system that incorporates different levels, both administrative and societal (Tänzler et al., 2013). Then, this study postulates that central actors, who hold the network together in times of distress, also have potential to influence adaptation information quality and flow in the network." Needs rephrasing. I suggest "Conflict-sensitive adaptation processes must be approached using a multi-dimensional system that incorporates different levels, both administrative and societal (Tänzler et al., 2013). This study postulates that central actors, who hold the network together in times of distress, also have potential to influence adaptation information quality and flow in the network." Accepted sentence flow change.

9. Page 16 – lines 501 – 502 and also other parts of the paper – We no longer have districts in Kenya but Counties and Sub-counties. The authors' needs to find out Loitoktok is now a sub-county. Accepted change from District to Sub-County officers (SCO) to correspond with current government administrative structures.